# Association between Gout and Dyslipidemia: A Nested Case–Control Study Using a National Health Screening Cohort

**DOI:** 10.3390/jpm12040605

**Published:** 2022-04-08

**Authors:** Hyo Geun Choi, Bong-Cheol Kwon, Mi Jung Kwon, Ji Hee Kim, Joo-Hee Kim, Bumjung Park, Jung Woo Lee

**Affiliations:** 1Department of Otorhinolaryngology-Head & Neck Surgery, Hallym University College of Medicine, Anyang 14068, Korea; pupen@naver.com (H.G.C.); pbj426@hallym.or.kr (B.P.); 2Hallym Data Science Laboratory, Hallym University College of Medicine, Anyang 14068, Korea; 3Department of Orthopedic Surgery, Hallym University College of Medicine, Anyang 14068, Korea; bckwon@hallym.or.kr; 4Department of Pathology, Hallym University College of Medicine, Anyang 14068, Korea; mulank99@hallym.or.kr; 5Department of Neurosurgery, Hallym University College of Medicine, Anyang 14068, Korea; kimjihee@hallym.or.kr; 6Division of Pulmonary, Allergy, and Critical Care Medicine, Department of Internal Medicine, Hallym University College of Medicine, Anyang 14068, Korea; luxjhee@hallym.or.kr; 7Department of Orthopedic Surgery, Yonsei University Wonju College of Medicine, Wonju 26426, Korea; 8Bigdata Platform Business Group, Wonju Yonsei Medical Center, Yonsei University, Wonju 26426, Korea

**Keywords:** gout, dyslipidemia, Korea, nested case-control, cholesterol

## Abstract

The association between lipid levels and uric acid disorders remains controversial. We evaluated the association between dyslipidemia and gout in a large cohort from the Korean National Health Insurance Service-Health Screening Cohort. Among the 514,866 participants aged ≥40 years, 16,679 gout participants were selected and matched with 66,716 control participants for income, region of residence, sex, and age. We used the ICD-10 codes to define dyslipidemia (E78) and gout (M10) and diagnosis was confirmed when each was reported ≥2 times. The odds ratios (ORs) of dyslipidemia history were calculated using conditional logistic regression in crude, partial, and fully adjusted models. The days of statin use, systolic and diastolic blood pressure, fasting glucose level, total cholesterol, obesity, Charlson comorbidity index, alcohol consumption, and smoking were used as covariates. Patients with gout had a significantly higher dyslipidemia history than those without gout (33.1% vs. 24.0%, *p* < 0.001). The association was significant after adjustment (OR in partial adjusted model = 1.50, 95% confidence interval (CI) = 1.44–1.57; OR in fully adjusted model = 1.43, 95% CI = 1.37–1.49). These findings were consistent with the subgroup analysis. Our findings suggest that dyslipidemia history is more likely in patients with gout aged ≥40 years than in healthy controls among Korean population.

## 1. Introduction

Characterized by a decrease in high-density lipoprotein (HDL) levels and an increase in low-density lipoprotein (LDL) and triglyceride (TG) levels, dyslipidemia increases with normal aging [1]. Small differences in total and non-HDL cholesterol values were reported globally over four decades [2]. Moreover, a declining trend in high cholesterol levels have been reported in the US [3]. A global analysis from 1980 to 2018 found a shift in lipid-related risk from high-income countries to other countries [2]. In a previous study, total or non-HDL cholesterol levels were elevated in East and Southeast Asia. In Korea, dyslipidemia diagnosis increased from 1.5 million to 11.6 million between 2002 and 2018 [4]. Therefore, dyslipidemia is a rather large public health problem in some countries.

Both the American guidelines in 2018 and European guidelines in 2019 recommended reducing LDL cholesterol [5]. Lipid disorders cause atherosclerosis and related clinical consequences such as peripheral arterial disease, sudden cardiac death, ischemic stroke, coronary heart disease, and heart failure [6]. Gout is a common inflammatory arthritis, and its prevalence and incidence are increasing globally [7]. Comorbidities are common in gout patients, and the European League Against Rheumatism recommends that every gout patient be evaluated for cardiovascular comorbidities and risk factors [8]. Therefore, lipid disorders and gout have similar comorbidities, and further studies are needed.

Many studies have analyzed the relationship between serum urate levels and lipid profiles, but the results remain controversial [9]. Previous studies have not analyzed the relationship between gout and dyslipidemia exclusively. Although previous studies have compared high uric acid levels, different values were not applied for men and women [9] and gout was not diagnosed [10,11,12,13,14,15,16]. Most studies have investigated the relationship between lipid profiles and uric acid levels, but there is a lack of recent data that analyze association of hyperlipidemia and gout using diagnostic codes. In addition, most previous studies did not consider comorbidities that can affect both diseases. Hence, additional cross-sectional studies with appropriate study designs are needed.

This study investigated the possibility of gout in patients with dyslipidemia using a nationwide population-based cohort sample. We hypothesized that a history of dyslipidemia influenced the consecutive development of gout. Comorbidities and medication history could affect the association between dyslipidemia and gout.

## 2. Materials and Methods

### 2.1. Study Population

The Hallym University Ethics Committee approved this study (2019-10-023). The Institutional Review Board waived the requirement for written informed consent as the analysis was based on secondary data. All analyses were performed in accordance with the guidelines and regulations of the Ethics Committee of Hallym University. We have described the Korean National Health Insurance Service-Health Screening Cohort (NHIS-HEALS) data in a previous study [17].

### 2.2. Definition of Gout

Gout was defined as patients who visited the clinic or hospital with a diagnosis of gout (ICD-10: M10) ≥ 2 times. The methods for diagnosis were modified from those reported in a previous study [18].

### 2.3. Definition of Dyslipidemia

Dyslipidemia was defined as ICD-10 code (E78) used before the day of gout treatment (index date). We assigned a dyslipidemia history when the participants were treated more than two times to ensure certainty of diagnosis.

### 2.4. Participant Selection

From a total of 514,866 participants with 615,488,428 medical claim codes, we selected 20,739 gout patients. We selected patients without gout from the total participants as our control group (n = 494,127). From the control group, we excluded participants without ICD-10 M10 codes from 2002 to 2015 (n = 10,214). To select patients diagnosed with gout for the first time, we excluded those diagnosed between 2002 and 2003 (washout periods, n = 4051). Participants without records of total cholesterol (n = 6), blood pressure (n = 2), and fasting blood glucose (n = 1) were excluded. We matched patients with control participants in a 1:4 ratio for age, sex, income, and region of residence. Control participants were chosen with random numbers to reduce selection bias. We matched the index date of each gout patient with the time of initiation of gout treatment and the index date of the control participants with the index date of their matched gout patients. Therefore, every gout patient and matched control participant had the same index date. During the matching process, 417,917 control participants were excluded. Finally, we matched 16,679 patients with gout and 66,716 control participants (Figure 1).

### 2.5. Covariates

To minimize the effect of possible confounders, such as blood pressure, days of statin use, comorbidity scores, cholesterol level, blood glucose level, alcohol consumption, smoking [19], and obesity, these were adjusted as covariates with different models.

Age groups were divided into five-year intervals: 40–44, 45–49, 50–54, … and 85+ years, and income groups were classified into 5 classes (class 1 (lowest income)–5 (highest income)). For the region of residence, we grouped participants as urban and rural residents according to our previous studies [17,20].

Before the index date, we collected data on tobacco smoking, alcohol consumption, and obesity using body mass index (BMI, kg/m^2^) [21] and the days of statin use for 2 years (730 days). The prescription dates of statin were counted for 2 years before the index dates for both the gout and control groups. Total cholesterol (mg/dL), systolic blood pressure (SBP, mmHg), diastolic blood pressure (DBP, mmHg), fasting blood glucose (mg/dL), and the Charlson comorbidity index (CCI) [22] were selected as covariates.

### 2.6. Statistical Analyses

We compared the general characteristics of the gout group and control groups using the chi-square test for categorical variables and independent *t*-test for continuous variables. Conditional logistic regression was used to analyze the odds ratio (OR) with 95% confidence intervals (CIs) for gout in patients with dyslipidemia and compared with those without dyslipidemia. We analyzed crude, partially adjusted (the days of statin use, SBP, DBP, total cholesterol, and fasting blood glucose), and fully adjusted (partially adjusted model and obesity, smoking, alcohol consumption, and CCI scores) models. The reports were stratified by age, sex, income, and region of residence. For the subgroup analyses, we divided participants by age (<60 years and ≥60 years old) and sex (men and women) and analyzed crude, model 1, and 2.

Additionally, we evaluated the subgroup analyses after adjusting for covariates (Appendix A). Finally, ORs with 95% CIs for gout and the days of statin use were calculated with subgroup analyses according to age and sex (Appendix A). Two-tailed analyses were used, and *p*-value < 0.05 was defined as significant. The SAS analytics software (version 9.4; SAS Institute Inc., Cary, NC, USA) was used for statistical analysis.

## 3. Results

The dyslipidemia and control participants had distinct general characteristics (Table 1). Patients with gout had a significantly higher (33.1%) history of dyslipidemia than those in the control group (24.0%) (*p* < 0.001).

Patients with gout had a significantly higher OR for previous hyperlipidemia when stratified by sex, age, region of residence, and income (*p* < 0.001, Table 2). The crude OR for hyperlipidemia was 1.59 (95% CI 1.54–1.66). Adjusted OR (aOR) maintained its significance in the partially adjusted model (aOR 1.50, 95% CI 1.44–1.57, *p* < 0.001) and the fully adjusted model (aOR 1.43, 95% CI 1.37–1.49, *p* < 0.001).

In the age and sex subgroup analysis, the findings were consistent in younger women and men of all ages when fully adjusted. The aORs were 1.51 (95% CI 1.41–1.63, *p* < 0.001) in men <60 years old, 1.47 (95% CI 1.26–1.71, *p* < 0.001) in women < 60 years old, and 1.42 (95% CI 1.32–1.51, *p* < 0.001) in men ≥60 years old. The aOR was not significant in the fully adjusted model for women ≥60 years old.

The aOR maintained its significance in the subgroup analysis adjusted for confounding factors such as fasting blood glucose, alcohol consumption, obesity, smoking, blood pressure, and total cholesterol (Appendix A).

The aOR for gout and the days of statin use/1 year maintained its significance in the fully adjusted model (aOR 1.07, 95% CI 1.03–1.11, Appendix A). In the subgroup analyses according to age and sex, only older women had a significant aOR.

## 4. Discussion

This study was based on the hypothesis that a history of dyslipidemia is associated with gout. We analyzed the association between dyslipidemia and the occurrence of gout using NHIS-HEALS data. Based on our results, patients with gout had a higher likelihood of having a history of dyslipidemia than participants in the control group. The findings were comparable, but inconsistent across the subgroups based on age and sex when adjusted for multiple variables.

Using the terms “hyperlipidemia”, “dyslipidemia”, “hyperuricemia”, and “gout”, we searched PubMed and Embase and defined our search for English articles before December 2021. There was no evidence that serum urate increased serum TG levels in a Mendelian randomization study [23]. Cross-sectional studies in the US [15], Italy [24], Korea [11], and Bangladesh [12] reported a relationship between lipid profiles and serum uric acid levels. In a Chinese study of 122,351 participants, increase in each TG level of 1 mmol/L had a higher OR (1.70–2.06) when serum urate levels were >420 μmol/L. The percentage of hypertriglyceridemia was 32.0% among patients with gout in a Taiwanese cohort [25], which was similar to the proportion of dyslipidemia (33.1%) in this study. The prevalence rate of high TG was 54.7% in 151 Korean patients with gout [26], but this high prevalence may be the effect of the inclusion criteria and study design. Gout had a high adjusted hazard ratio (HR, 1.40; 95% CI 1.31–1.50) for hyperlipidemia in the UK database [27]. In a recent Taiwanese study, the risk of hyperlipidemia was higher in patients with gout than in healthy controls (HR 2.55, 95% CI 2.50–2.61), and the HR was higher in men than in women [28]. Unlike previous similar studies, this study utilized extensive representative public data with sufficient statistical power, it involved analyses after adjustment for many covariates that may be related to gout and dyslipidemia, and it had a 1:4 matched case–control study design.

According to previous studies, serum uric acid levels are significantly associated with a history of lipid disorders. The pooled OR for dyslipidemia was 1.84 (95% CI 1.49–2.28) for the highest versus lowest level of uric acid in a meta-analysis [10]. Qi et al. analyzed the association between serum urate levels and lipid profiles in 122,351 participants [9]. The ORs for a high serum urate level (>420 µmol/L) were significantly elevated with increasing TG levels. Confounders may have influenced the results of the crude analysis; therefore, adjustment could have decreased the effect of confounding factors. In the US general population, gout is associated with an increased prevalence of Type 2 diabetes mellitus (26%) and hypertension (74%) [29]. In the same study, individuals with hyperuricemia had a higher OR (3.12, 95% CI 2.43–4.01) for obesity than those without hyperuricemia [29]. In addition, heavy alcohol consumption (HR 1.38, 95% CI 1.28–1.49) and former smokers (HR 1.14, 95% CI 1.06–1.22) showed a higher risk of incident gout than the controls in a cohort study [30]. Unlike previous studies that investigated gout and hyperlipidemia, the strength of our study is that several covariates were adjusted during analysis.

Gout and dyslipidemia share pathophysiological mechanisms including aging, comorbidities, and diet. The risk of gout increases with age [7], and total cholesterol and LDL-C levels increase, while HDL-C levels decrease with age [1]. In addition, patients with gout have a greater burden of comorbidities than the general population. In a UK study, gout was associated with comorbidities; aOR increased from 1.39 to 2.51, as the CCI increased from 1–2 to ≥5 [27]. Fructose is closely associated with hyperuricemia [31]. Fructose intake leads to fatty acid accumulation and dyslipidemia [32], while adenosine triphosphate (ATP) depletion in the liver results in a decreased intracellular phosphate [31]. Further depletion of phosphate causes overproduction of urate from purine nucleotide catabolism. As fructose consumption has progressively increased worldwide, further studies on the relationship between fructose intake and uric acid levels are required.

Statin use was considered as an adjustment factor to address confounding effects. Further, we analyzed the ORs for gout with the days of statin use/1 year because the treatment duration can influence the outcome. In Korea, the incidence of new statin users has increased [33], and high statin maintenance doses are likely to increase Type 2 diabetes cases [34]. A rat study showed that simvastatin blocks L-type Ca^2+^ and consequently inhibits insulin excretion, glucose-induced Ca^2+^ concentration, and signaling [35]. Moreover, previous studies have proposed a decline in insulin resistance and secretion due to statin [36]. Insulin resistance can be induced by fatty acids; high concentrations of fatty acids increased reactive oxygen and nitrogen species, leading to oxidative stress in tissues [37]. Our study showed a significant but slight increase (aOR 1.07, CI 1.03–1.11) in gout with increased use of statin/1 year in a full adjustment model. Thus, the accumulated effects of statin use can result in uric acid disorders.

Contrary to our results, statin therapy significantly reduced plasma uric acid levels in a meta-analysis [38]. However, the results differed according to statin type; it was statistically significant for atorvastatin and simvastatin but not for rosuvastatin, pravastatin, and lovastatin. However, differences in lipid-lowering drugs used according to country, race, and study design may have influenced the results. Therefore, further studies are required to evaluate this aspect.

Our study had several limitations. First, although we used objective criteria such as ICD-10 codes to diagnose dyslipidemia and gout, the components of dyslipidemia could not be thoroughly evaluated. Hyperlipidemia is generally defined as high LDL, total cholesterol, TG levels, lipoprotein levels (≥90th percentile), or low HDL levels (≤10th percentile) [39]. However, we utilized extensive representative public data with sufficient statistical power. Further studies with detailed laboratory results are required. Second, we analyzed middle-aged and older people. The outcomes may differ between younger age groups and other ethnic groups. Third, although we adjusted for several confounders, there may be unmeasured confounders such as anthropometric measurements, marital status, education level, diet, physical exercise, history of diabetes, and hypertension. However, our study included more variables than previous studies and is a 1:4 matched case–control study. In addition, the NHIS-HEALS is national screening data, and thus selection bias is less likely, especially when the control group was selected randomly. Fourth, we could not identify causality, owing to the cross-sectional nature of the study. The large population has statistical power, but a prospective longitudinal study would provide a better explanation of the relationship between dyslipidemia and gout.

## 5. Conclusions

This study investigated the possibility of gout in patients with dyslipidemia using a nationwide population-based cohort sample. When matched for age, sex, income, and region of residence, patients with gout were more likely to have a previous history of dyslipidemia than healthy controls. This study revealed a link between gout and dyslipidemia, and the results support the detrimental effect of lipid disorder on inflammatory arthritis.

## Figures and Tables

**Figure 1 jpm-12-00605-f001:**
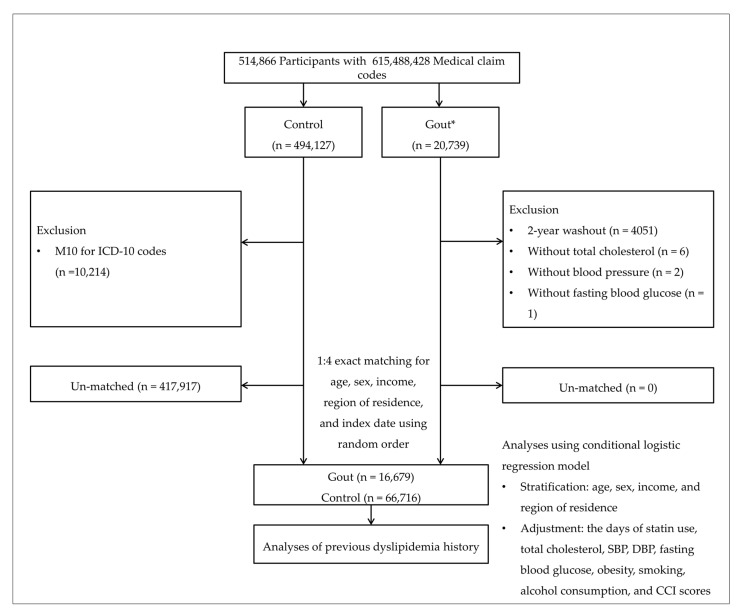
A schematic illustration of the participant selection process. Of 514,866 participants, 16,679 gout participants were matched with 66,716 control participants for age, sex, income, and region of residence. DBP, diastolic blood pressure; SBP, systolic blood pressure; * Gout: Gout was selected when the participant was assigned M10 based on ICD-10 codes ≥ 2 times.

**Table 1 jpm-12-00605-t001:** Characteristics of all participants.

Characteristics	Total Participants		
	Gout	Control	*p*-Value
Age (years old, n, %)			1.000
40–44	355 (2.1)	1420 (2.1)	
45–49	1743 (10.5)	6972 (10.5)	
50–54	3171 (19.0)	12,684 (19.0)	
55–59	3141 (18.8)	12,564 (18.8)	
60–64	2623 (15.7)	10,492 (15.7)	
65–69	2320 (13.9)	9280 (13.9)	
70–74	1772 (10.6)	7088 (10.6)	
75–79	1061 (6.4)	4244 (6.4)	
80–84	403 (2.4)	1612 (2.4)	
85+	90 (0.5)	360 (0.5)	
Sex (n, %)			1.000
Male	13,278 (79.6)	53,112 (79.6)	
Female	3401 (20.4)	13,604 (20.4)	
Income (n, %)			1.000
1 (lowest)	2354 (14.1)	9416 (14.1)	
2	2084 (12.5)	8336 (12.5)	
3	2551 (15.3)	10,204 (15.3)	
4	3509 (21.0)	14,036 (21.0)	
5 (highest)	6181 (37.1)	24,724 (37.1)	
Region of residence (n, %)			1.000
Urban	7091 (42.5)	28,364 (42.5)	
Rural	9588 (57.5)	38,352 (57.5)	
Total cholesterol (mg/dL, mean, SD)	199.8 (40.2)	196.5 (37.7)	<0.001 ^†^
SBP (mmHg)	129.7 (17.1)	127.4 (16.4)	<0.001 ^†^
DBP (mmHg)	80.5 (11.1)	79.1 (10.7)	<0.001 ^†^
Fasting blood glucose (mg/dL)	101.7 (28.1)	102.1 (31.7)	0.118
Obesity (n, %) ^‡^			<0.001 *
Underweight	218 (1.3)	1667 (2.5)	
Normal	4237 (25.4)	23,280 (34.9)	
Overweight	4585 (27.5)	18,681 (28.0)	
Obesity grade I	6948 (41.7)	21,485 (32.2)	
Obesity grade II	691 (4.1)	1603 (2.4)	
Smoking status (n, %)			<0.001 *
Non-smoker	9545 (57.2)	37,431 (56.1)	
Past smoker	3376 (20.2)	12,330 (18.5)	
Current smoker	3758 (22.5)	16,955 (25.4)	
Alcohol consumption (n, %)			<0.001 *
<1 time a week	8665 (52.0)	37,811 (56.7)	
≥1 time a week	8014 (48.1)	28,905 (43.3)	
CCI score (score, n, %)			<0.001 *
0	10,486 (62.9)	45,574 (68.3)	
1	2578 (15.5)	9111 (13.7)	
2	1558 (9.3)	5289 (7.9)	
3	854 (5.1)	2873 (4.3)	
≥ 4	1203 (7.2)	3869 (5.8)	
The days of statin use (day, mean, SD)	79.4 (195.5)	57.0 (169.7)	<0.001 ^†^
Dyslipidemia (n, %)	5523 (33.1)	16,034 (24.0)	<0.001 *

Abbreviations: CCI, Charlson comorbidity index; DBP, diastolic blood pressure; SBP, systolic blood pressure. * Chi-square test. Significance at *p* < 0.05; ^†^ Independent *t*-test. Significance at *p* < 0.05; ^‡^ Obesity (BMI, body mass index, kg/m^2^) was categorized as <18.5 (underweight), ≥18.5 to <23 (normal), ≥23 to <25 (overweight), ≥25 to <30 (obesity grade I), and ≥30 kg/m^2^ (obesity grade II).

**Table 2 jpm-12-00605-t002:** Gout in dyslipidemia and non-dyslipidemia groups with subgroup analyses according to age and sex.

Characteristics	Odds Ratios for Gout					
	Crude ^†^	*p*-Value	Partial ^†,‡^	*p*-Value	Full ^†,§^	*p*-Value
Total participants (n = 83,395)						
Dyslipidemia	1.59 (1.54–1.66)	<0.001 *	1.50 (1.44–1.57)	<0.001 *	1.43 (1.37–1.49)	<0.001 *
Non-dyslipidemia	1		1		1	
Age < 60 years, men (n = 34,370)						
Dyslipidemia	1.74 (1.64–1.85)	<0.001 *	1.62 (1.51–1.74)	<0.001 *	1.51 (1.41–1.63)	<0.001 *
Non-dyslipidemia	1		1		1	
Age < 60 years, women (n = 7680)						
Dyslipidemia	1.56 (1.37–1.78)	<0.001 *	1.49 (1.28–1.74)	<0.001 *	1.47 (1.26–1.71)	<0.001 *
Non-dyslipidemia	1		1		1	
Age ≥ 60 years, men (n = 32,020)						
Dyslipidemia	1.54 (1.46–1.64)	<0.001 *	1.49 (1.40–1.60)	<0.001 *	1.42 (1.32–1.51)	<0.001 *
Non-dyslipidemia	1		1		1	
Age ≥ 60 years old, women (n = 9325)						
Dyslipidemia	1.41 (1.27–1.56)	<0.001 *	1.25 (1.11–1.40)	<0.001 *	1.23 (1.09–1.39)	0.001
Non-dyslipidemia	1		1		1	

Abbreviations: CCI, Charlson comorbidity index; DBP, diastolic blood pressure; SBP, systolic blood pressure. * Conditional logistic regression, Significance at *p* < 0.05; ^†^ Models were stratified by age, sex, income, and region of residence; ^‡^ Partially adjusted model: Adjusted for the days of statin use, total cholesterol, SBP, DBP, and fasting blood glucose; ^§^ Fully adjusted model: Adjusted for the days of statin use, total cholesterol, SBP, DBP, fasting blood glucose, obesity, smoking, alcohol consumption, and CCI scores.

## Data Availability

The data included in this study are available from NHIS-HEALS, but restrictions apply to availability. These data were used under a license for this specific study and are not publicly available.

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
