# Peer review of "Association between Gout and Dyslipidemia: A Nested Case–Control Study Using a National Health Screening Cohort"

_jpm, 2022, doi:10.3390/jpm12040605_

Round 1

Reviewer 1 Report

This is a well-presented article about the association between dyslipidemia and gout in Korean population. Strengths and limitations of the study are provided in the Discussion. It is not clear though what is the novelty of the current study in comparison to other studies related to gout and dyslipidemia in Korean subjects (e.g. ref 22).

Figure 1 is missing from the manuscript file.

Author Response

Answer) Thank you for appreciating our manuscript and for the comments. In response to your comment, we have described the novelty of the current study in comparison to other studies in the discussion. We utilized extensive representative public data with a sufficient statistical power. Also, we adjusted many covariates that may be related to gout and dyslipidemia and conducted a 1:4 matched case-control study. This information has been documented in the Discussion section.

We apologize for this omission. We have placed Figure 1 where it is cited in the manuscript.

Lines 225–228: Unlike previous similar studies, this study utilized extensive representative public data with sufficient statistical power, involved analyses after adjustment for many covariates that may be related to gout and dyslipidemia, and had a 1:4 matched case-control study design.

Reviewer 2 Report

The manuscript is well-written and the topic is of interest to the readers.

The following aspects require confirmation / further explanation:

  1. Previous research have investigated the link between uric acid level and the occurrence of hypertension, ischemic heart diseases, and heart failure. This manuscript aims to explore the correlation between gout and hyperlipidemia. However the conceptual flow of thinking in the introduction is too subtle / unclear. Please revise.
  2. Please add more information about the urgency of this topic in the introduction.

Author Response

Answer) Thank you for your comments. We agree with you that the introduction was not satisfactory; therefore, we revised the Introduction section by reinforcing the flow and moving some information to the Discussion section.

Lines 75–76: Therefore, lipid disorders and gout have similar comorbidities, and further studies are needed.

Lines 78–81: Previous studies have not analyzed the relationship between gout and dyslipidemia exclusively. Although previous studies have compared high uric acid levels, different values were not applied for men and women [9] and gout was not diagnosed [10-16].

Lines 83–85: In addition, most previous studies did not consider comorbidities that can affect both diseases. Hence, additional cross-sectional studies with appropriate study designs are needed.
